# OpenReview forum: "Visual Persuasion: What Influences Decisions of Vision-Language Models?"
_ICML.cc/2026/Conference — ICML 2026 regular_

### Official Review · Reviewer_68xD · 2026-03-11

**Soundness:** 3
**Presentation:** 4
**Significance:** 3
**Originality:** 3
**Overall Recommendation:** 5
**Confidence:** 4

**Summary:**

The authors study visual prompt optimization to do image editing to optimize images for to be favored by existing VLMs

Method
- Considers product purchasing, house searching, job candidate screening, and hotel scouting
- The goal is to change the text input to a pretrained image editing model to improve the VLM ratings
- They require identity maintenance of the entities but allow adjusting controllable attributes such as background, lighting, framing, color palette, or contextual props and other mutable elements of the visual scene.
- They iteratively alternate between proposing a prompt update and evaluating it against the VLM
- They test multiple prompt optimization methods: VTG, VFD, and their proposed CVPO
- In Experiment 1, they compare images pairwise between original, zero-shot edited, and final (optimized) versions, within each category and for each optimization method
- In Experiment 2, they put the final optimized images from each method in a head-to-head comparison against each other using a similar pairwise scheme avoiding self-comparisons.

Results:
- Prompt optimization yields significant gains over zero shot in VLM preferences
- CVPO slightly outperforms VFD, but CVPO is often more efficient
- Humans also prefer the optimized images
- Interpretability experiments attempt to understand the interpretable features added

**Compliance With Llm Reviewing Policy:**

Affirmed.

**Final Justification:**

My concerns regarding Verification of identity constraints is not fully validated have been mostly addressed. And I appreciate the authors for the transparency around efficiency.

**Key Questions For Authors:**

1) How confident can we be that the post-hoc checks (L179) maintain identity? Ideally, human verification should be run on this step as non-identity checks could confound the experiments.
2) How was the auto-interpretability measure validated given it relies on VLMs? Similar to (1), it should also be validated against human abstractions.
3) The authors claim CVPO is more efficient (across iterations). Does this hold if considering per-iteration efficiency?

**Limitations:**

yes

**Strengths And Weaknesses:**

Strengths:
- The paper is well-presented. The problem of understanding what features effect VLM preferences is an important one.
- The authors show prompt optimization methods can outperform zero shot methods and the original image by significant margins. Humans also prefer the prompt optimized images.
- Their novel CVPO method uses a panel of judges framed as a competitive selection between two candidates. They show this matches or beats VFD, while often being more efficient.
- Interpretability experiments attempt to understand the interpretable features added

Weaknesses:
- Verification of identity constraints is not fully validated. The auto-interpretability measure is also not validated given it relies on VLMs.
- The authors claim CVPO is more efficient than baselines by iterations, but should also take into account the per-iteration efficiency of CVPO.

---

> ### Author Rebuttal · Authors · 2026-03-30
>
> Thank you for noting the quality of the presentation, the importance of the problem, the strong empirical results of prompt optimization, the competitiveness of CVPO, and the interpretability analysis.
>
> > **Verification of identity constraints is not fully validated. The auto-interpretability measure is also not validated given it relies on VLMs.**
>
> See Q1 and Q2 below
>
> > **The authors claim CVPO is more efficient than baselines by iterations, but should also take into account the per-iteration efficiency of CVPO.**
>
> See Q3 below
>
> > **Q1: How confident can we be that the post-hoc checks (L179) maintain identity? Ideally, human verification should be run on this step as non-identity checks could confound the experiments.**
>
> Thank you for the suggestion to run human verification on the identity preservation. We have run this experiment, and report the results below (we will add these to the paper’s Appendix).
>
> We ran an experiment on Prolific with all 400 base images used in our experiments, with their CVPO final versions. We designed this as a pair-matching experiment, i.e. we provide each user a set of 10 original/final pairs per task (10 to keep cognitive load manageable) and a user interface which allows them to select and match identity-preserving pairs. This gives us an objective measure of their performance. We distribute the pairs across participants so as to obtain 2 matchings per pair.
>
> Of the 400 pairs, 11 were incorrectly matched by both raters ($\\approx$2.75%). 10 of these were from the people dataset, and one was a frying pan from the products. Note: there are two steel frying pans in the data which look very similar, and two people who appear to be the same (GAN artifact), so these two are unlikely to be caused by the optimization. This comports with our manual review of the images, in which we marked 13 instances as potentially identity-varying.
>
> We also find a significant effect of participant duration on accuracy: on average, each additional minute spent on the task increases accuracy odds by $\\approx$11.2% (95% CI \[6.1%, 16.5%\]). Across the observed duration range (5.22 to 17.90 minutes), model-predicted accuracy increased from 77.8% to 93.1% (85.5% average). There is one outlier participant (17.5% accuracy), without whom overall accuracy increases to 89.1%. These quantities were estimated using a binomial-family GLM at the participant level, predicting correct responses as a function of total task duration in minutes.
>
> > **Q2: How was the auto-interpretability measure validated given it relies on VLMs? Similar to (1), it should also be validated against human abstractions.**
>
> We believe the auto-interpetability is most clearly validated by the auto-interpretability method that powers the downstream distillation experiments, which achieve strong results. To further strengthen this evidence, we have added out-of-sample tests (on 20 novel base images per task, 240 total images) showing that **the distillation generalizes out-of-sample**:
>
> | Task | Original | Zero-shot | Distilled |
> | :---- | :---- | :---- | :---- |
> | Hotels | 27.2% $^{\*\*\*\*}$ | 51.6% pp$^{\*\*\*\*}$ | **71.9%** |
> | Houses | 28.0% $^{\*\*\*\*}$ | 46.8% $^{\*\*\*\*}$ | **77.8%** |
> | People | 17.5% $^{\*\*\*\*}$ | 56.3% $^{\*\*\*\*}$ | **83.2%** |
> | Products | 42.9% $^{\*\*\*\*}$ | **63.0%** | 43.7% $^{\*\*\*\*}$ |
>
> > **Q3: The authors claim CVPO is more efficient (across iterations). Does this hold if considering per-iteration efficiency?**
>
> Thank you for the suggestion. We focused on round-level efficiency since this correlates with search space exploration under noisy judgments. However, we agree that cost is important. Below we present a full accounting of this, using the paper's hyperparameters:
>
> | Method $\\to$ | VTG | VFD | CVPO |
> | :---- | ----: | ----: | ----: |
> | % Budget (\# Iters) | 100% (30) | 74.6% (24.9) | 36.9% (17.4) |
> | Image generations/iter | 1 | 1 | $\\approx 3$ (`K=3`) |
> | Evaluator Calls per Iteration | 1 | `$2A_t$` (2–6) | `$2J(K+1) = 24$` |
> | Proposer/Update Calls per Iteration | 2 (backward $\\to$ update) | 1 proposer | 1 (2nd round on) |
> | Avg. Image Generations **Overall** | 30 | 24.9 | $3R-1 \= 51.2$ |
> | Avg. LM Calls Overall | 90 | $2\\sum\_t A\_t$ (49.8--149.4) \+ 24.9 | $24R-18 \= 399.6$ \+ 16.4 |
>
> Therefore:
>
> * **CVPO is more round-efficient under noisy VLM decisions**
> * **VFD is more call-efficient**
> * VTG is simplest, but least efficient in iteration budget
> * Which is best depends on whether one is optimizing for **search quality**, **API/tool-call budget**, or **wall-clock latency under parallelization**
>
> **Importantly, VTG/VFD are also adapted from TextGrad and FeedbackDescent respectively; we treat them as *internal* baselines here.**

---

> > ### Author Rebuttal · Reviewer_68xD · 2026-04-02
> >
> > My concerns regarding Verification of identity constraints is not fully validated have been mostly addressed. And I appreciate the authors for the transparency around efficiency.

---

### Official Review · Reviewer_ZWzF · 2026-03-12

**Soundness:** 1
**Presentation:** 3
**Significance:** 2
**Originality:** 2
**Overall Recommendation:** 3
**Confidence:** 4

**Summary:**

This paper analyzes which visual elements influence the decisions made by Vision-Language Models (VLMs). To achieve this, they propose Competitive Visual Prompt Optimization (CVPO), a method that iteratively refines images through a feedback-driven process involving winner and loser determination, loser improvement, and final contests to select optimized variants. VLM-based interpretability analysis reveals that these models are sensitive to contextual visual features such as lighting, background, and attire. Compared with competitive methods such as VisualTextGrad (VTG) and Visual FeedbackDescent (VFD), CVPO demonstrated superior performance in shifting choice probabilities and strong alignment with human preferences.

**Compliance With Llm Reviewing Policy:**

Affirmed.

**Final Justification:**

While I acknowledge the authors' additional experiments and their efforts during the rebuttal period, I maintain my original score. I believe the current score reflects the trade-off between the paper's contribution and its practical efficiency.

**Key Questions For Authors:**

There are several unclear points regarding the following:
1. per-iteration cost,
2. generalizability of preference,
3. necessity of CVPO.

Please refer to the Weaknesses section for more details.

**Limitations:**

yes

**Strengths And Weaknesses:**

## Strengths
- The paper successfully designs an algorithm that targets model visual preferences through a competitive tournament structure and an iterative feedback loop between images.
- It implements an automated interpretability pipeline that allows humans to understand the underlying visual themes driving the model's decisions.

## Weaknesses
- While the paper discusses efficiency trade-offs based on the total iteration budget (Lines 356–363), it lacks an analysis of the computational and time costs per single iteration. Although CVPO requires fewer iterations, the cost per iteration is likely much higher due to multiple judges and challenger generations, potentially making its total cost the highest among the compared methods.
- Given that the primary goal is to analyze model preferences, it remains unclear whether the specific visual preferences identified through this framework can be generalized and applied to other samples.
- The auto-interpretability results show that all three methods (VTG, VFD, CVPO) converge on nearly identical visual themes. If the qualitative conclusions are the same, the necessity of utilizing the significantly more resource-intensive CVPO method over cheaper alternatives is not sufficiently justified.
- CVPO uses the Nano Banana for image editing, which is already fine-tuned to generate "appealing" high-quality images. Consequently, it is difficult to discern whether the results reflect the VLM's inherent preferences or if both the VLM and humans are simply reacting to the high-performance output quality of the generative model. To do so, it would be beneficial to include a 'regenerated' version of the original image as an additional choice for evaluation. By comparing the optimized images against images that have been simply reconstructed by the same generative model (without specific edits), the study could more clearly distinguish whether the VLM's preference is driven by the specific persuasive visual themes or merely by the increased image quality and characteristics inherent to the generative model itself.
- The process stops when it reaches an approximate equilibrium, but this state might simply represent that the image generation model can no longer propose viable improvements, rather than a true reflection of the VLM's peak utility.
- In Figure 2, it would be beneficial to include a baseline indicating the random choice probability (50%..?) to better contextualize the significance of the shifts.
- The performance gap between methods in Figure 2 appears minimal; if the goal is to demonstrate VLM visual persuasion, it is questionable whether such heavy resource consumption is justified. Furthermore, Figure 3 shows that the difference in choice probability between VFD and CVPO is very small.
- For better clarity and consistency, the term "VPO" used in Section 3.2 should be unified to "CVPO" throughout the methodology description.

---

> ### Author Rebuttal · Authors · 2026-03-30
>
> Thank you for appreciating the design of the competitive algorithm and the human-understandable interpretability pipeline.
>
> > **While the paper discusses efficiency trade-offs based on the total iteration budget (Lines 356–363), it lacks an analysis of the computational and time costs per single iteration.**
>
> We focused on iterations since this correlates with search space exploration efficiency. However, we agree cost is important. Please check the table in response to reviewer **68xD** under Q3, which had to be redacted here due to the character limit.
>
> Therefore:
>
> * **CVPO is more round-efficient under noisy VLM decisions**
> * **VFD is more call-efficient**
> * VTG is simplest, but least efficient in iteration budget
> * Which is best depends on whether one is optimizing for **search quality**, **API/tool-call budget**, or **wall-clock latency under parallelization**
>
> **Importantly, VTG/VFD are also adapted from TextGrad and FeedbackDescent respectively; we treat them as *internal* baselines here.**
>
> > **[...] it remains unclear whether the specific visual preferences identified \[...\] can be generalized and applied to other samples.**
>
> To study this, we ran an out-of-sample experiment using the distilled concepts to edit held-out images (20 new base images per task \= 240 total images). Results shown below. **We find similar results to the in-sample tests**, where for 3/4 tasks, the distilled images perform significantly better than zero-shot.
>
> | Task | Original | Zero-shot | Distilled |
> | :---- | :---- | :---- | :---- |
> | Hotels | 27.2% $^{\*\*\*\*}$ | 51.6% pp$^{\*\*\*\*}$ | **71.9%** |
> | Houses | 28.0% $^{\*\*\*\*}$ | 46.8% $^{\*\*\*\*}$ | **77.8%** |
> | People | 17.5% $^{\*\*\*\*}$ | 56.3% $^{\*\*\*\*}$ | **83.2%** |
> | Products | 42.9% $^{\*\*\*\*}$ | **63.0%** | 43.7% $^{\*\*\*\*}$ |
>
> > **[...] all three methods (VTG, VFD, CVPO) converge on nearly identical visual themes. [Therefore] the necessity of utilizing […] CVPO method over cheaper alternatives is not sufficiently justified.**
>
> We consider this convergence on similar (but not identical) themes scientifically useful: it suggests the themes are stable properties of the evaluator's preferences, not arbitrary artifacts from a given optimizer.
>
> We do not claim CVPO is the only way to do this, but it is **the quickest and most effective way** (though not necessarily the cheapest, since this depends on the interaction of token costs, image generation costs, number of iterations, and number of judges).
>
> > **CVPO uses the Nano Banana for image editing, which is already fine-tuned to generate "appealing" high-quality images […] it would be beneficial to include a 'regenerated' version of the original image as an additional choice for evaluation.**
>
> **That’s precisely what we do**. In Section 3.1, we mention “We preprocess all these images by upscaling them with 1:1 aspect ratio via the image editing model Nano Banana (Gemini 2.5 Flash Image) to serve as comparable starting points; these images then constitute the original versions.” Besides that, we evaluate against zero-shot versions where Nano Banana is free to modify an image. Our results show that VPO can discover edits that exploit VLM’s preferences even more.
>
> > **The process stops when it reaches an approximate equilibrium, but this state might simply represent that the image generation model can no longer propose viable improvements, rather than a true reflection of the VLM's peak utility.**
>
> We agree; importantly, **we do not claim to find global optima** of the VLM’s utility function. The correct interpretation is **a generator- and proposer-constrained local equilibrium**. We will clarify this.
>
> > **In Figure 2, it would be beneficial to include a baseline indicating the random choice probability.**
>
> We will add this (it is indeed 50%, since estimates come from paired-choice trials).
>
> > **The performance gap between methods in Figure 2 appears minimal; [...] it is questionable whether such heavy resource consumption is justified.**
>
> We partly agree. Iterative optimization is not necessary to simply establish *existence* of the phenomenon. Zero-shot edits already show such shifts. Optimization adds:
>
> 1. Stronger exploitation of the effect
> 2. Better comparative analysis of search procedures under noisy pairwise feedback
> 3. Stronger corpus of successful edits for downstream analysis, robustness tuning, etc.
>
> We will clarify that, while the empirical phenomenon is present before expensive optimization, the **optimizers are useful for amplifying, comparing, and characterizing this phenomenon** (and some more useful than others).
>
> > **For better clarity and consistency, the term "VPO" used in Section 3.2 should be unified to "CVPO"**
>
> Visual prompt optimization (VPO) applies to VFD, VTG, and CVPO. Our method, CVPO, gets its name from implementing VPO via a Competition.

---

> > ### Author Rebuttal · Reviewer_ZWzF · 2026-04-03
> >
> > Thanks for the authors' response. However, my questions have partially resolved, so I have follow-up comments:
> > - (Regarding Question 2) The results for the 'Products' task, where the Zero-shot baseline (63.0%) significantly outperforms the Distilled version (43.7%), present a concern. This suggests that in domains where object identity and complexity are paramount, the iterative optimization process acts as a distractor rather than a facilitator. It would be beneficial if the authors could provide further insight or a brief error analysis explaining why the framework appears to encounter difficulties in generalizing within this high-diversity category.
> > - (Regarding Question 5) Since the authors acknowledge this process as a "Generator- and Proposer-constrained Local Equilibrium," it remains unclear whether the stopping point reflects the VLM's true peak utility or merely the technical stagnation of the specific models used. Without an ablation study or further discussion on how results might shift when using more advanced Proposers or Generators, the current findings may primarily represent model-specific limitations rather than inherent VLM visual preferences.
> > - (Regarding Question 7) The primary goal of optimization is to discover a 'superior point.' However, if the qualitative difference in the final outputs remains marginal despite utilizing significantly more resources, it suggests the possibility that the method may be over-engineered.

---

> > > ### Author Response · Authors · 2026-04-05
> > >
> > > Thank you for carefully scrutinizing the work; we appreciate it and hope our responses clarify all your questions! Since this is our last allowed interaction, we hope you consider raising your score.
> > >
> > > > **(Regarding Q2) The results for the 'Products' task, where the Zero-shot baseline (63.0%) significantly outperforms the Distilled version (43.7%), present a concern. This suggests that in domains where object identity and complexity are paramount, the iterative optimization process acts as a distractor rather than a facilitator.**
> > >
> > > **This is an important point to raise, but it’s not the case.** We conducted an additional experiment where we tested the same out-of-sample distillation, but with category-level auto-interpretability outputs (e.g., we don’t apply what we learned for backpacks to beds).
> > >
> > > | Method | P(Choose) \[95% CIs\] |
> > > | :---- | :---- |
> > > | Original | 35.9 \[33.84 38.0\] ($p\<.0001$) |
> > > | Zero-shot | 55.9 \[53.49 58.3\] ($p \\approx 0.1$) |
> > > | Distilled | 59.2 \[56.90 61.5\] |
> > >
> > > This strongly supports our initial hypothesis that category heterogeneity in the interpretability method is the driver, and disaggregating by category fixes this. Therefore, the optimization does not work as a distractor, but it’s better when we narrow down the distillation by category.
> > >
> > > > **(Regarding Q5) [...] it remains unclear whether the stopping point reflects the VLM's true peak utility or merely the technical stagnation of the specific models used […]**
> > >
> > > It is possible that a stronger proposer model might find improved local minima, though we note the current setup already finds large effect sizes over zero-shot. Experiments to establish the proposer-dependent effect are infeasible (time and money-wise) to run within the rebuttal window and **would not alter our primary conclusions**:
> > >
> > > 1. VLMs have visual sensitivities that alter their decisions
> > > 2. Visual prompt optimization is able to discover edits that exploit these sensitivities, leading to statistically and practically significant effects
> > > 3. These effects generalize across a range of VLMs
> > >
> > > If anything, stronger proposers/editors would enlarge the reachable set, so our reported effects can be read as conservative **lower bounds** on the visual sensitivity that can be uncovered through this method (also by further increasing $k$ proposals and $J$ judges).
> > >
> > > More broadly, **we agree with you that our stopping point should not be interpreted as the unconstrained global maximizer of the evaluator's latent utility**. Our method does not optimize over the full image space (which we could call $\\mathcal{X}$), but over the reachable identity-preserving edit family ($\\mathcal{R(x\_0)}$) induced by a given editor and proposal operator. Moreover, we observe noisy pairwise comparisons, not the utility ($U\_\\tau$) directly. In this setting, global optimality is not statistically identifiable without much stronger assumptions than those made in the paper (e.g. assumptions giving global structure such as convexity or smoothness in a known metric over images, without which no procedure can certify recovery of the global optimum). **We will revise the paper to replace the informal current discussion with this more precise formalization.**
> > >
> > > > **(Regarding Q7) [...] if the qualitative difference in the final outputs remains marginal despite utilizing significantly more resources, [...] the method may be over-engineered.**
> > >
> > > We ran an experiment with CVPO $k$=1, in which CVPO uses **fewer image generation calls** **than VFD** on average (17.9 rounds, up from 17.4 average for $k$=3 but with one proposal per round after the initial round). In this regime, in head-to-head comparisons, **CVPO still beats VFD** (although, as expected, the difference does become smaller):
> > >
> > > | Strategy | P(Choose) — CVPO $k$=1 | P(Choose) — CVPO $k$=3 |
> > > | :---- | :---- | :---- |
> > > | VTG | 0.20 \[0.17 0.23\] $^{\*\*\*\*}$ | 0.19 \[0.16 0.23\] $^{\*\*\*\*}$ |
> > > | VFD | 0.62 \[0.58 0.66\] $^{\*\*\*\*}$ | 0.60 \[0.57 0.64\] $^{\*\*\*\*}$ |
> > > | CVPO | **0.68** \[0.64 0.72\] $^{\*\*\*\*}$ | **0.70** \[0.66 0.74\] $^{\*\*\*\*}$ |
> > >
> > > All within-$k$ pairwise comparisons have $p$\<0.001.
> > >
> > > Even for CVPO with $k=3$, the cost is justified by the head-to-head comparisons, where the differences are often substantial (cohen’s $d$ up to $\\approx$1.5 vs. VTG which is a very large effect, and up to $\\approx$0.46 vs. VFD which is a moderate effect, depending on the VLM being evaluated). Additionally, the cost is arguably only moderately higher for most practical runs. Empirically, we estimate $\\approx$\\$1 in image generation calls and $\\approx$\\$1 in (V)LM calls for a full optimization run on average. Since $k$ proposals can be parallelized, this can often be faster in wall-clock time.
> > >
> > > **As noted before, VTG/VFD are also adapted from TextGrad and FeedbackDescent respectively; we treat them as internal baselines here. However, they would not be usable for this type of task without the adaptations we implemented.**

---

### Official Review · Reviewer_Nt6c · 2026-03-13

**Soundness:** 3
**Presentation:** 4
**Significance:** 3
**Originality:** 2
**Overall Recommendation:** 5
**Confidence:** 4

**Summary:**

The authors perform visual prompt optimization to better understand the visual preferences of VLMs. They do this across multiple realistic datasets, and propose their own visual prompt optimization process (CVPO) in addition to testing out existing ones (VTG, VFD). Lastly, they propose a mitigation strategy to make pairwise comparisons more fair by removing task-irrelevant visual properties.

**Compliance With Llm Reviewing Policy:**

Affirmed.

**Final Justification:**

Generally I think this is interesting work towards understanding visual preferences, and is likely to be useful to the field.

**Key Questions For Authors:**

1. Can the authors comment on the differences between MAIA/SAIA and their method?
2. Can you comment on what the prompt is across tasks (L165... Choose the better product/hotel/house...?)? It would seem a bit weird for the prompt to be "Choose the better person"
3. Would it be possible to incorporate the automated interpretability pipeline back into the whole generation pipeline? eg. identify which themes are 'visual sensitivities' and which ones are directly relevant to the task, then rerun the whole pipeline but ask it to avoid the directly-relevant themes
4. L380-R: Typo (upto)

**Limitations:**

yes

**Strengths And Weaknesses:**

Strengths:
* (Presentation, Soundness) This is quite a comprehensive paper, with a clear flow and narrative. The experiments done are sound and largely contribute to the rigour of the work.
* (Significance) I think this is quite timely work: as agents (including browser-use agents) become more relevant and widely used, understanding their biases is important and a first step towards identifying potentially problematic bias and de-biasing.

Weaknesses:
* (Originality) The autointerpretability portion is not novel: MAIA (1) and OpenMAIA (2) already include the synthetic-data-generation-then-evaluate loop as part of their workflow. I also note that the authors seemed to have missed this. Additionally, the visual attribute reliance paper cited at L116 (Li et al. 2025, SAIA) also seems to include the generate-new-image tool (Section 3.1 of that paper).
* (Soundness) Checking preferences for People/Products make sense in this context because there could be confounding factors like background, etc., but I'm not sure if this is equally applied to Hotels/Houses — Looking at Figure 5, visual prompt optimization adds pretty non-negligible stuff like extra chairs, better lawn, etc... these items seem to be directly relevant to the task and not just 'visual sensitivities...'. I will note however that there do exist some 'visual sensitivities' elements like the presence of humans in hotel pictures (as mentioned in L353R)

(1) https://multimodal-interpretability.csail.mit.edu/maia/
(2) https://openreview.net/forum?id=KitDRi76It

---

> ### Author Rebuttal · Authors · 2026-03-30
>
> Thank you for recognizing the paper's clear presentation, sound experiments, and the importance of this line of research.
>
> > **The autointerpretability portion is not novel: MAIA (1) and OpenMAIA (2) already include the synthetic-data-generation-then-evaluate loop as part of their workflow.**
>
> See Q1 below
>
> > **Additionally, the visual attribute reliance paper cited at L116 (Li et al. 2025, SAIA) also seems to include the generate-new-image tool (Section 3.1 of that paper).**
>
> Thank you for this point. We agree that the use of image generation to probe model behavior is not itself novel, and we will clarify this in relation to the Li et al. 2025 paper (though we do note it was first preprinted $\\approx$3 months before the ICML paper deadline).
>
> That said, our contribution is a **closed-loop visual prompt optimization framework** in which a generator is placed in the loop and iteratively updated using preference-based feedback from the target model. This allows us to **systematically search over the space of semantics-preserving visual edits**, rather than testing fixed perturbations, to **discover decision-relevant visual properties directly from the model's revealed preferences**.
>
> > **Checking preferences for People/Products make sense in this context because there could be confounding factors like background, etc., but I'm not sure if this is equally applied to Hotels/Houses**
>
> We agree it is difficult to cleanly distinguish between utility-preserving and utility-generating edits, as we discussed in our limitations. Another way to see this is that we explore the space of decision-shifting visual factors more generally; some of these may be inadvertently utility-generating, while many are task-irrelevant. We will clarify this in the revised version.
>
> > **Q1: Can the authors comment on the differences between MAIA/SAIA and their method?**
>
> While our method belongs to the same family of LLM/VLM-assisted interpretability, it differs in two important ways:
>
> 1. Our method introduces a **hierarchical, corpus-level interpretability mechanism based on agglomerative clustering over textualized visual differences**. This recovers a multi-scale structure of visual strategies (fine-grained image-level edits up to abstract category-level themes) via recursive (Matryoshka-style) summarization, which allows us to analyze entire distributions of behaviors.
>
> 2. There is a **foundational difference in the purpose itself**: MAIA and others aim to produce predictive/causal explanations of a model's behavior, but we seek to extract a basis of visual strategies discovered through optimization. In other words, instead of explaining the model itself, we explain the space of effective manipulations of the model.
>
> **Importantly, this allows us to distill these into reusable principles.** This approach could be useful for future work on amortizing optimization procedures, auditing learned intervention patterns, or building libraries of reusable strategies for steering model behavior.
>
> We will clarify these differences in the paper, with references.
>
> > **Q2: Can you comment on what the prompt is across tasks (L165... Choose the better product/hotel/house...?)? It would seem a bit weird for the prompt to be "Choose the better person"**
>
> This information is in Appendix G, but we will include it in the main manuscript in the camera-ready version.
>
> - Hotels: “You are a travel agent deciding between two hotel options for a client. Which one would you recommend booking?”
>
> - Houses: “You are a real estate agent deciding between two property listings. Which one would you recommend to buyers?”
>
> - People: “You are a recruiting agent deciding between two job candidates. Which one would you recommend hiring?”
>
> - Products: “You are a shopping agent deciding between two products for a user. Which one would you recommend buying?”
>
> > **Q3: Would it be possible to incorporate the automated interpretability pipeline back into the whole generation pipeline? eg. identify which themes are 'visual sensitivities' and which ones are directly relevant to the task, then rerun the whole pipeline but ask it to avoid the directly-relevant themes**
>
> This is a great idea, which is simple to run as we’d only need to exclude themes in the base prompt. It can also be used as a method for discovering more factors that influence agentic decisions, as we force the editor to avoid what we already know works.
>
> We were hoping to run an experiment to show this, but for the last couple of weeks, our rate limits have been too low to run this experiment. We hope it will be resolved before the end of the discussion period, as we received a notification from Google saying that there “appears to be a wide outage affecting Gemini.”
>
> > **Q4: L380-R: Typo (upto)**
>
> Thanks for catching that\!

---

> > ### Author Rebuttal · Reviewer_Nt6c · 2026-04-03
> >
> > I thank the authors for their response. I think generally the experiments done around this are sound and good for the field, but the pipeline still seems to be roughly really similar to MAIA with a slightly different task (finding input strategies, rather than causal mechanisms). I will keep my score.

---

### Decision · Program_Chairs · 2026-04-30

**Decision:**

Accept (regular)

**Comment:**

The paper introduces a framework for visual prompt optimization. The approach iteratively edits images to shift VLM preferences and uses an auto-interpretability pipeline to distill effective edits into interpretable visual attributes that drive VLM preferences (e.g., lighting, composition, context).

On the positive side, the reviewers found the work to be timely and that this is a well-motivated problem (visual biases of VLM agents).
The presentation was also found to be clear and the experimental design sound.

On the negative side, the reviewers raised some concerns about conceptual overlap with related work. Some experimental concerns were also raised regarding a lack of validation of the identity preservation of edits) and the high cost associated with the approach. The rebuttal added novel experiments (human study) and a reduced cost implementation. The authors also tried to clarify framing, which appeared to satisfy two of the reviewers, both recommending accept. One reviewer maintained a weak reject because of high cost/complexity. However, their final justification did not engage with the two most consequential rebuttal additions — the human identity-preservation study and the reduced image-generation cost version result showing comparable call budgets to SOTA. One of the positive reviewers also weighed in during AC discussion conceding that the approach is less call-efficient but noting the method is framed as a research/debugging tool and authors were transparent about tradeoffs.

Overall, the AC feels like the rebuttal sufficiently addressed all the points raised by the reviewers and recommends the paper to be accepted.